# The Impact of miRNA in Colorectal Cancer Progression and Its Liver Metastases

**DOI:** 10.3390/ijms19123711

**Published:** 2018-11-22

**Authors:** Ovidiu Balacescu, Daniel Sur, Calin Cainap, Simona Visan, Daniel Cruceriu, Roberta Manzat-Saplacan, Mihai-Stefan Muresan, Loredana Balacescu, Cosmin Lisencu, Alexandru Irimie

**Affiliations:** 1Department of Functional Genomics, Proteomics and Experimental Pathology, The Oncology Institute “Prof. Dr. Ion Chiricuta”, Cluj-Napoca 400015, Romania; simona.visan19@gmail.com (S.V.); dani_cruceriu@yahoo.com (D.C.); loredana_balacescu@yahoo.com (L.B.); 211th Department of Medical Oncology, University of Medicine and Pharmacy “Iuliu Hatieganu”, Cluj-Napoca 400015, Romania; dr.geni@yahoo.co.uk (D.S.); calincainap2015@gmail.com (C.C.); 3Department of Medical Oncology, The Oncology Institute “Prof. Dr. Ion Chiricuta”, Cluj-Napoca 400015, Romania; 4Department of Molecular Biology and Biotechnology, “Babes-Bolyai” University, Cluj-Napoca 400006, Romania; 55th Department of Internal Medicine, University of Medicine and Pharmacy “Iuliu Hatieganu”, Cluj-Napoca 400006, Romania; roberta786@yahoo.com; 6Department of Gastroenterology, Wilhelmina Hospital Assen, Europaweg-Zuid 1, 9401 RK Assen, The Netherlands; 7Department of Surgery, Municipal Hospital, University of Medicine and Pharmacy “Iuliu Hatieganu”, Cluj-Napoca 400139, Romania; mihai.stefan.muresan@gmail.com; 811th Department of Oncological Surgery and Gynecological Oncology, University of Medicine and Pharmacy “Iuliu Hatieganu”, Cluj-Napoca 400015, Romania; cosminlisencu@yahoo.com (C.L.); airimie@umfcluj.ro (A.I.); 9Department of Surgery, The Oncology Institute “Prof. Dr. Ion Chiricuta”, Cluj-Napoca 400015, Romania

**Keywords:** miRNAs, colorectal cancer, exosomes, liver, metastases

## Abstract

Colorectal cancer (CRC) is one of the most commonly diagnosed malignancies with a high incidence and mortality rate. An essential challenge in colorectal cancer management is to identify new prognostic factors that could better estimate the evolution and treatment responses of this disease. Considering their role in cancer development, progression and metastasis, miRNAs have become an important class of molecules suitable for cancer biomarkers discovery. We performed a systematic search of studies investigating the role of miRNAs in colorectal progression and liver metastasis published until October 2018. In this review, we present up-to-date information regarding the specific microRNAs involved in CRC development, considering their roles in alteration of Wnt/βcatenin, EGFR, TGFβ and TP53 signaling pathways. We also emphasize the role of miRNAs in controlling the epithelial–mesenchymal transition of CRC cells, a process responsible for liver metastasis in a circulating tumor cell-dependent manner. Furthermore, we discuss the role of miRNAs transported by CRC-derived exosomes in mediating liver metastases, by preparing the secondary pre-metastatic niche and in inducing liver carcinogenesis in a Dicer-dependent manner.

## 1. Introduction

Colorectal cancer (CRC) is the third most frequently diagnosed cancer worldwide, with an annual incidence of 1.4 million new cases and 694,000 deaths [1]. About 15% of CRCs are diagnosed in metastatic stages (stage IV), with an average survival rate of 2.5 years. Moreover, about 50% of patients that have a low TNM stage at diagnosis will eventually develop metastasis, mostly localized in the liver [2]. During the last decade, the clinical outcome of patients with metastatic CRC has improved due to the development of new chemotherapeutic drugs and targeted therapies. Today, the median overall survival for patients with metastatic disease is about 30 months, time that has doubled over the past 20 years [3]. 

Despite the current screening methods and prognosis factors (TNM stage, age, tumor differentiation grade, vessel invasion, performance status and tumoral markers according to the National Comprehensive Cancer Network (NCCN) guidelines), there are still a great number of patients that are experiencing therapeutic failure and metastasis [4]. Therefore, an important task in CRC management is to identify new prognostic factors that could help in selecting patients who could benefit from adjuvant therapies or from intensive screening for disease recurrence and metastasis.

After their discovery, micro-RNAs (miRNAs) have been shown to have important implications in cancer biology. Increasing evidence highlights that deregulated miRNAs’ expression has a functional role in the progression and metastasis of CRC, acting either as tumor suppressors or oncogenes to regulate expression of their specific mRNA targets. Due to their high stability, miRNAs were considered and investigated as a new class of valuable biomarkers [5].

We will present herein up-to-date information about the specific miRNAs involved in CRC progression and liver metastasis and how these miRNAs control the epithelial-mesenchymal transition (EMT) of CRC cells, a process responsible for liver metastasis in a circulating tumor cell-dependent manner. We also emphasize the role of miRNAs transported by CRC-derived exosomes in mediating liver metastases, by preparing the secondary pre-metastatic niche and in inducing liver carcinogenesis in a Dicer-dependent manner.

## 2. Selection of the Studies to Review

We performed a systematic search of studies profiling the miRNAs involved in colorectal progression and liver metastasis, published until October 2018. Four PubMed string searches (Figure 1) were used for this aim; in the first search we identified the miRNAs involved in CRC progression and metastasis, considering their roles in alteration of Wnt/βcatenin, EGFR, TGFβ and TP53 signaling pathways; in the second search we selected the miRNAs involved in the EMT, an important step in CRC metastasis; in the third, we identified the specific miRNAs transported by tumor-delivered exosomes (miRNA-TEX) released by CRC, which support the liver metastasis by preparing the secondary pre-metastatic niche, inducing pro-inflammation and immunosuppression; in the fourth, we found out the overlapping between miRNAs-TEX and miRNAs involved in promoting the hepatocellular carcinoma, in a Dicer-dependent manner.

## 3. The Role of the Tumor Microenvironment in CRC Development and Progression

In 2000, Hanahan and Weinberg [6] stated that there are “six essential alterations that collectively dictate malignant growth: self-sufficiency in growth signals, insensitivity to growth-inhibitory (antigrowth) signals, evasion of programmed cell death (apoptosis), limitless replicative potential, sustained angiogenesis and tissue invasion and metastasis”. A decade later, when tumors were recognized as organs, compared to what was previously accepted as solitary masses of proliferating cells, an enhancement of these cancer hallmarks was provided. Thus, reprogramming energy metabolism and evading immune response became the newest hallmarks of cancers while tumor microenvironment (TME) has been proved to play a substantial role in tumor progression and metastasis [7]. A TME represents a very complex network between tumor cells and stromal, endothelial and immune cells, which substantially contributes to the validation of an aggressive tumor phenotype [8]. Moreover, the presence of inflammatory cells and inflammatory mediators such as chemokines and cytokines related to TME facilitates tumor progression, including CRC [9]. Besides that, TME facilitates CRC progression by maintaining a paracrine cross-talk signaling between tumor resident adipocytes that represent a fuel rich source of energy and cancer cells demanding energy for their rapid proliferation [10]. 

Nowadays, it is largely accepted that CRC occurs from an adenoma–carcinoma transition and implies genetic and epigenetic events, supported by TME [11]. The genetic and epigenetic studies in CRC have provided a lot of data about the mutational status and alteration of gene expression [12]. So far, a plethora of papers have broadly presented all these alterations that lead to “oncogene addiction” by activating several key oncogenes (e.g., *KRAS*, *EGFR*, *MYC*, *ERBB2*, etc.) or inactivating tumor suppressor genes (e.g., *TP53*, *PTEN*, *WNT*, *SMAD2*, *SMAD4*, etc.) and DNA repair genes (e.g., *bMSH2*, *bMLH1*, *bPMS1*, *bPMS2* and *MSH6*) [11,13,14]. 

Colorectal carcinogenesis is also mediated by epigenetic alterations, including changes in the DNA methylation status, both by global hypo-methylation and promoter hyper-methylation. Global hypomethylation decreased by 10–40% in CRC compared to normal colonic tissue, being associated with genomic instability, while promoter hyper-methylation leads to repression of several tumor suppressor genes and their specific pathways (e.g., *TP53*, *PI3K*/*PTEN*/*AKT*/*mTOR*, *TGFb*/*SMAD*) [15]. Furthermore, non-coding RNAs, especially miRNAs, play important roles in maintaining the physiology of normal colonic cells, while alteration of their levels contributes to CRC development, progression and metastasis, as well as drug resistance and tumor relapse [16]. Moreover, recent evidence indicates that miRNAs are involved in both direct cell-to-cell signaling and paracrine signaling between tumor cells and TME components as secreted molecules in microvesicles or exosomes [17]. 

## 4. A Brief Overview of miRNAs

Micro-RNA (miRNA, miR) represent the most studied class of non-coding RNAs, being responsible for negative modulating of up to 60% of protein-coding genes expression [18]. One of the most important features of miRNAs consists of their multi-target potential, such as a single miRNA can target up to 200 mRNAs, while different miRNAs can modulate the same mRNA target [19]. The biogenesis of miRNAs has been extensively presented before [20]. Shortly, the biogenesis of miRNA starts in the nucleus, with transcription of a long hairpin transcript (pri-miRNA) of hundreds or thousands of nucleotides. Further, by an enzymatic process coordinated by RNA polymerase III Drosha and DGCR8 (DiGeorge syndrome critical region 8) pri-miRNA is reduced to a smaller transcript of about 70 nucleotides, called pre-miRNA. After it is exported in the cytoplasm by nuclear receptor exportin 5, pre-miRNA is firstly reduced by Dicer complex to a mature miRNA duplex of about 22 nucleotides lengths and then to a single-stranded mature miRNA. Further, mature miRNA-loaded AGO2 and RNA-induced silencing complex (RISC) will function as a guide to target specific mRNA transcripts by sequence complementarity, usually in the 3′ untranslated region (UTR), leading to translational repression or mRNA degradation (Figure 2). 

Croce’s group [21] demonstrated for the first time that miRNAs alteration is responsible for cancer development and also they reinforced the hypothesis that miRNA acts as the negative regulators of their target mRNAs [22]. Immediately afterward a worldwide research effort has been taken to identify and characterize new cancer-related miRNAs. Currently, the latest data from miRNA database miRBase Release 22 (http://www.mirbase.org/) include 2815 mature human miRNAs. Nevertheless, a big challenge consists of identification of the cancer-specific mechanisms altered by miRNA modulation. Consequently, a large number of miRNA:mRNA target prediction tools, included different algorithms, have been developed. The question about which is the best algorithm does not yet have a precise answer, although TargetScan represents one of the most considered options [23]. However, accuracy prediction of the miRNA:mRNA interaction is still challenging and the regulatory function of each miRNA should to be experimentally validated, specific for each pathology. The functionality of oncogenes and tumor suppressor genes in cancer is tightly controlled by two classes of miRNAs, called tumor-suppressor miRNAs (TS-miRNAs, TS-miRs) and oncomiRs, respectively. Genetic and epigenetic alterations that underlie cancer development lead to “loss of function” or inactivation of TS-miRs and “gain of function” or over-activation of oncomiRs. As a result, the oncogenes, that are the targets of TS-miR, become overexpressed, while the tumor suppressor genes are inhibited by the overexpression of the oncomiRs [24]. 

## 5. The Role of miRNAs in CRC Progression and Metastasis

The first association of miRNAs with colon cancer was described by Michael et al. [25] fifteen years ago when they reported decreased miR-143 and miR-145 expression in colon cancer when compared to normal colon tissue. Since then, many studies have been conducted to investigate the miRNAs involvement in colorectal carcinogenesis. Considering the evolutionary progression of CRC from adenomas, in-depth understanding of the miRNAs-modulated molecular mechanisms in CRC has become a major challenge [26,27]. It has been shown that alterations in Wnt/βcatenin, EGFR, TGFβ and TP53 signaling pathways result in survival, proliferation, invasion and metastasis of CRC [28,29]. Therefore, many studies have focused on these pathways, to establish the miRNAs:mRNA interaction networks, in order to complete the molecular puzzle that underlines the CRC development, progression and metastasis. Adenomatous polyposis coli (*APC*) is an important tumor suppressor gene with an inhibitory function on Wnt/ß-catenin signaling. When mutated, as happens in 90% of CRC, *APC* loses its inhibitory function, leading to tumorigenesis, invasion and metastasis. An increasing number of studies have identified specific miRNAs that lead to alteration of APC/Wnt/ß-catenin signaling, either by direct suppression of *APC* (eg miR135a/b, miR-494, miR-19a) and aberrant activation of the Wnt pathway (eg miR-21, miR-155, miR-103a, miR-1827, miR-145, miR-34a), or indirectly, targeting other members of these pathways (miR-150, miR-224, miR-146a, miR-574-5p). An up-to-date list of oncomiRs and tumor suppressor miRNAs (TS-miRNAs), and their targets involved in the alteration of Wnt/ß-catenin signaling in CRC is presented in Table 1.

The EGFR signaling pathway has a critical role in normal embryogenesis, mediating cell proliferation, differentiation, migration and apoptosis. The gain of function of *EGFR* by genetic alterations leads to cancer development. About 50% of CRC present *EGFR* gene amplification and mutational activation of *KRAS* and *BRAF* downstream mediators [71]. Recent studies have reported that aberrant activation of oncogenic EGFR pathway can be due to TS-miRNAs loss of function. Two of the most important TS-miRNAs involved in the EGFR pathway are miR-143 and miR-145, whose combined action decreases proliferation and migration by targeting several members of the EGFR pathway, including *KRAS* and *BRAF* [49]. Loss of function of let-7a was associated with high levels of *KRAS* and *c-MYC* and colon tumorigenesis [72], while the let-7 *KRAS* rs712 polymorphism was correlated with increased colorectal cancer risk [50]. Recent evidence has confirmed that miR-19a can inhibit CRC angiogenesis by targeting *KRAS* and *VEGFA* [51] and that miR-181d reduces cell proliferation, migration and invasion by triggering *PEAK1*, a downstream regulator in EGFR/KRAS pathway [52]. On the other hand, activation of the KRAS pathway can trigger and upregulate some oncomiRs, such as miR-210 and miR-181a with a role in CRC development and progression [73] or miR200c that has a pivotal role in CRC aggressiveness and metastasis [74]. 

The AKT-PI3K-mTOR pathway represents the second signaling hub of the EGFR pathway, being amplified in about 15–20% CRC [75]. Amplification of the AKT-PIK3K-PTEN pathway is mediated both by the loss of function of TS-miRs, such as miR-126, miR-497 and miR-1, or by upregulation of several oncomiRs including miR-21, miR-19 and miR-96 [76]. More data about miRNAs involved in the EGFR signaling pathway in CRC developing and progression are presented in Table 1, section EGFR signaling pathway.

TGF-ß signaling has a dual function in colorectal carcinogenesis. It is a tumor suppressor in early stages of CRC development, but acts as a tumor promoter in advanced stages of CRC. The tumorigenic activity of TGF-ß signaling is due to several up- and downstream defects that enhance multiple oncogenic pathways leading to increased angiogenesis, evasion of the immune system, evading apoptosis, EMT and metastasis [77]. Mutational alterations of *TGFBR2*, the key component that initiates signaling in the TGF-β pathway, were identified in about 30% of CRCs. However, recent evidence indicates that many miRNAs are involved in *TGFBR2* regulation (Table 1). Several of them, including miR-301a and miR-135b, induce proliferation, migration and invasion in CRC cells through negative regulation of *TGFBR2* expression [59,60]. Also by negative regulation of *SMAD4*, a mediator of the TGF-β pathway, miR-20-5p and mir-224, induces EMT, invasion and metastasis of CRC cells [61,62]. 

*TP53*, “the guardian of the genome”, is a tumor suppressor gene with multiple roles in maintaining the cellular physiology under many stress conditions, including cancer [78]. Almost 50% of CRCs present inactivation of *TP53* by deletions or point mutations [79]. Albeit *TP53* can affect transcription and maturation of many miRNAs, both by transactivation of tumor-suppressor miRNAs and by repression of oncogenic miRNAs, there is increasing evidence that *TP53* expression is also under the tight control of miRNAs [80]. Translational repression of *TP53* in CRC is controlled by several miRNAs including miR-125b, miR-504, miR-25, miR-30d and miR-638. Previous data have demonstrated that miR-125b is an independent prognostic factor in CRC, its high expression being associated with poor prognosis [63], while mir-504 was shown to have a role in the negative regulation of *TP53* in several cancers, including CRC [64]. Moreover, miR-25b and miR-30 were found to reduce apoptosis by negative regulation of both gene expression and protein level of *TP53* [65]. Interestingly, miR-518c and miR-638 can target and inactivate both *TP53* and *PTEN* genes [67]. However, several studies have reported that miR-638 can also function as tumor suppressor miRNAs, its loss of function leading to proliferation, EMT, migration and invasion of CRC by upregulating SOX2 and TSPAN1 proteins [68,69]. Another study has reported that *TP53* can indirectly be repressed by *SIRT1*, a target of tumor suppressor miR-34a [70]. 

However, miRNAs-mRNAs networks in CRC are more complex in the more specific case of CRC metastasis. MiRNAs can control the EMT underlying the movement of the colorectal circulating tumor cells (CTCs). By their encapsulation in exosomes, miRNAs provide support in preparing the liver metastatic niche, suppressing of the immune system, and modulating the liver phenotype in a Dicer-dependent manner.

## 6. The Role of miRNAs in CRC Metastasis, by Modulation of EMT

Understanding the molecular mechanisms underlying CRC metastasis is crucial for improving the treatment strategies for this pathology and consequently, to increase the patients’ survival rate. Although the miRNAs-mediated mechanisms in colorectal carcinogenesis are quite well understood, those responsible for metastasis are not yet elucidated.

An important feature of cancer cells that lead to invasion and metastasis consists in changing their epithelial characteristics to mesenchymal ones, a process known as the epithelial to mesenchymal transition (EMT). During EMT, cancer cells undergo several processes that modify their phenotype, leading to cell motility, the acquisition of stemness properties, inhibition of apoptosis and immunosuppression. Epithelial to mesenchymal transition is characterized by dissolution of epithelial tight cell junctions by downregulation of several specific genes that encode proteins like E-cadherin (*CDH1*) and claudins, and by promoting the mesenchymal adhesion by activating several protein-coding genes, as the ones for vimentin (*VIM*), N-cadherin (*NCAD*) and fibronectin (*FN1*). Stimuli from TME such as inflammatory cytokines and growth factors have a significant role in tumor evolution, by abnormally regulating the EMT of cancer cells [81]. The EMT modulation requires a complex network of cooperation including many regulators and inducer molecules [82]. 

The main regulators of EMT are represented by three families of activating transcription factors: *SNAIL*, *ZEB* and *TWIST*. However, accumulating evidence has shown that two other families of transcription factors, *PROX1* and *FOX*, are involved in EMT of CRC. Nevertheless, the induction of EMT by members of these families of transcription factors involves multiple pathways, tightly coordinated by miRNAs (Table 2). 

MiR-34 family (miR-34a/b/c), directly regulated by *TP53*, is involved in preventing *TGFβ*-induced EMT by suppressing *SNAIL*, *SLUG* and *ZEB1* transcription factors, as well as the stemness factors *BMI1*, *CD44, CD133*, or *c-MYC*. Conversely, *SNAIL* and *ZEB1* repress miR-34a/b/c expression, promoting EMT. However, the loss of function of *TP53* and/or miR-34a/b/c, found in many cancers, represents an important molecular alteration facilitating cancer metastasis [84]. Epithelial to mesenchymal transition activation by *SNAIL* in CRC directly induces *ZNF281* transcription, leading to repression of miR-34a/b/c that contributes to CRC metastasis [85]. 

In a recent study [86], miR-375 was proved to regulate *MMP2* and several EMT-associated genes, including *SNAIL*. When it is downregulated, miR-375 leads to proliferation, invasion and migration of colorectal cancer cells. MiR-374 represents another tumor suppressor, whose loss of function in CRC induces the activation of the PIK3/AKT pathway, promoting proliferation, invasion, migration and intrahepatic metastasis. Transcription factors *SNAIL*, *SLUG* and *ZEB1* as well as *NCAD* and *VIM* are among the targets of miR-374, all of which being significantly upregulated by miR-374 inhibition [87]. MiR-200c and miR-429, two members of the miR-200 family, are predominately involved in the regulation of *ZEB* transcription factors in CRC cells. Induction of miR-200c leads to inactivation of EMT by suppressing *ZEB1* expression which results in reduced invasion and migration of CRC cells [88]. MiR-429 could reverse *TGFβ*-induced EMT by targeting *ONECUT2*, and thus, inhibiting cell migration and invasion. However, miR-429 is significantly downregulated in CRC [90]. On the other hand, activation of miR-200 cluster by loss of function of *ASCL2*, a downstream target of WNT signaling, leads to inhibition of *ZEB* and *SNAIL* families of transcription factors, and consequently, to the regulation of the plasticity from EMT to mesenchymal-epithelial transition (MET) [89]. Downregulation of other tumor suppressors, mainly miR-335, miR-132 and miR-192 was related to invasion and metastasis of CRC by increasing expression of their *ZEB*2 target gene [91,92,93]. In hypoxic conditions, miR-675-5p overexpression can regulate hypoxia-induced EMT driven by *HIF1α*, through increasing *SNAIL* transcription by both inhibiting the *SNAIL*’s repressor *DDB2* and stabilizing the activity of the transcription factor *HIF1α* [83].

Although a few papers have described several miRNAs targeting *PROX1*, none of them has focused on CRC. However, the role of *PROX1* inducing EMT in CRC was previously pointed out. *PROX1* can promote EMT by inhibiting *CDH1* expression via binding to the promoter of pre-miR-9-2 and triggering its expression [109]. On the other hand, two members of the FOX family of transcription factors, *FOXQ1* and *FOXM1*, have been recently identified as targets of miR-320c, as part of the EMT induction process. Vishnubalaji R et al. [101] reported low levels of miR-320 in CRC compared to normal tissues and an inverse correlation with *SOX4*, *FOXM1* and *FOXQ1* expression, leading to decreased *CDH1* expression. 

The main inducers of EMT in CRC are TGF-ß and Wnt/ß-catenin pathways but *TMPRSS4*, *FMNL2*, *GDF15*, *NRP2* or *TUSC3* were also linked to EMT in CRC [82]. TGF-ß-induced EMT in CRC is related to both *SMAD*-dependent and -independent mechanisms and partly influenced by alteration of miRNAs expression. The role of *SMAD4* in EMT has been extensively studied. Previous studies have demonstrated that normal levels of *SMAD4* maintain the epithelial phenotype while inactivation of *SMAD4* leads to invasion and metastasis. Downregulation of mir-187, a downstream effector of the TGF-β pathway with a role in EMT prevention, leads to increased expression of *SOX4*, *NT5E* and *PTK6*, known as important upstream regulators of the *SMAD* pathway. Loss of function of miR-187 was tightly correlated with CRC metastasis and a reserved prognosis [102]. On the other hand, over-expression of miR-20a induces EMT, promoting metastasis of CRC via suppression of *SMAD4* expression [94]. Moreover, the *TGFß*-induced EMT can be initiated by suppressing the *SMAD* inhibitors, such as *SMAD7*. Thereby, miR-4775, miR-1269 and miR-21 promote CRC metastasis in a *SMAD7/TGFβ* signaling-dependent manner [96,97,98]. MiR-150, the most expressed miRNA induced by Wnt/ß-catenin pathway, enhances the EMT in CRC by targeting the cAMP response element-binding protein (CREB) signaling pathway [99]. 

Although Wnt-induced EMT in CRC is mostly activated by canonical Wnt/ß-catenin signaling, there is evidence that also non-canonical Wnt signaling is involved in EMT. The Wnt-induced EMT could be activated by suppressing the negative regulators of several transcription factors. Accordingly, loss of function of the tumor-suppressors miR-34a, miR-145 and miR-29b leads to the induction of EMT. Mir-34 is a suppressor of the ß-catenin pathway and EMT [85,103] but its inhibition by methylation leads to cancer progression and metastasis [104]. Likewise, the loss of function of miR-145 was inverse correlated with the overexpression of several genes involved in EMT, such as *CTNNB1*, *VIM* and *SNAI*, and the downregulation of *CDH1* expression in SW480 cells [105]. The regulation of EMT in colon cancer cells via miR-29b involves blocking of various β-catenin target genes, by suppressing the coactivators of β-catenin such as *TCF7L2*, *SNAIL* and *BCL9L* [106]. Moreover, recent data has shown that miR-29b-3p modulated by *lncRNAH19* can promote the EMT in CRC by directly binding and targeting the progranulin (*PGRN*) [107]. 

Recent evidence has shown that circulating tumor cells (CTCs) from CRC are responsible for liver metastasis [110], while the EMT-MET plasticity of CTCs is considered an important hallmark of metastasis [111]. Both the detachment of CTCs from the primary CRC and their attachment in the liver are closely mediated by miRNA. Circulating tumor cells detachment is facilitated by increasing the activity of the matrix metalloproteinases (*MMPs*) or by inhibiting the tissue inhibitors of metalloproteinase (*TIMPs*). On this regard, miR-194 can promote EMT-mediated metastasis of CRC by *MMP2* activation [100], while miR-20a promotes CRC metastasis by inhibiting *TIMP2*, that leads to increasing of *MMP2* and *MMP9* enzymatic activity [95]. Moreover, by destabilizing *MMP11* mRNA, let-7c plays a role as a metastasis suppressor, in addition to its tumor growth suppressor activity [108]. While CTCs are responsible for tumor metastasis, miRNAs-TEX are responsible for preparing the secondary metastatic niche.

## 7. The Role of miRNAs-TEX in Sustaining the CRC Liver Metastasis

The occurrence of liver metastases from CRC represents a multistep process that involves EMT-MET plasticity of colorectal CTCs, local remodeling of the liver microenvironment and immunosuppression. Generally, these processes are strongly mediated by miRNAs-TEX released by CRC. 

Exosomes are small extracellular vesicles of about 30–100 nm surrounded by a lipid bilayer, derived from multivesicular bodies (MVBs) generated within the cell’s endosomal system. They are secreted by multiple cell types, including tumor cells, and can be detected in most body fluids such as breast milk, urine, saliva, semen and blood, as well as in supernatants of cultured cells. Exosomes can modulate the phenotype of the cells they bind to by transferring them to their cargo content, either in a paracrine or endocrine manner [112]. Preparing the secondary pre-metastatic niche or remodeling the liver microenvironment is a mandatory condition to sustain liver metastasis. Exosomes released by CRC can transfer bioactive molecules including nucleic acids (DNA, mRNA, miRNA, lncRNA, circRNA etc), proteins (transcription factors, receptors, enzymes, etc), and lipids to the liver cells, modulating their phenotype [113]. One of the most important features of TEXs is that they can determine the organotropism and predict organ-specific metastasis due to their distinct integrin expression patterns. Integrins bind to specific receptors found on target cells, preparing the secondary pre-metastatic niche by inducing the pro-inflammatory signals through Src phosphorylation and S100 gene expression [114]. Two chemokine receptors, *CXCR4* [115] and *CCR6* [116] have been previously associated with CRC metastasis to the liver (Figure 3). Moreover, previous data exploring the characteristics of the metastatic pattern of CRC has revealed that the liver is the main metastatic site of CRC, while lungs, bones and the brain are less common [117].

According to the existing data, miRNAs are the most studied exosomes-entrapped molecules that are involved in cancer development, progression and metastasis. Since 2010, when the functional role of miRNAs-TEX in gene expression regulation in recipient cells was demonstrated [118], their essential contribution to the hallmarks of cancer has been acknowledged [119]. However, the amounts of miRNAs that are incorporated into the exosomes released by different tumors remain to be elucidated. One hypothesis mentions that the abundance of miRNAs-TEX depends on the presence of their mRNA target transcripts in the cytoplasm of the donor cell. Therefore a presumptive mRNA-miRNA interaction in the donor cell could lead to low levels of that miRNA in the exosomes [112]. On the other hand, miRNAs sorting into exosomes could be conditioned by their posttranscriptional modifications, so that 3′ end adenylated miRNAs are preponderantly enriched in cells while 3′ end uridylated miRNAs are enriched in exosomes [120]. 

Wang et al. [121] demonstrated for the first time that exosomes play a pivotal role in colorectal cancer liver metastasis. Using nude mouse models, the authors demonstrated that exosomes derived from a highly liver metastatic CRC cell line (HT-29) can induce the same aggressive liver metastasis in the Caco-2 xenograft mouse model that usually has poor liver metastatic potential. The mechanisms by which HT-29-derived exosomes influence the liver metastasis of CRC involve increased *CXCR4* expression in many types of stromal cells that contribute to the remodeling of the liver microenvironment and thereby to the development of the secondary pre-metastatic niche. Moreover, the treatment of Caco-2 cells line with HT-29-derived exosomes has led to increased migration of Caco-2 cells, highlighting the main role of exosomes in colorectal liver metastasis. Since then, several studies have identified miRNAs-TEX as relevant molecules in CRC liver metastasis. 

Accordingly, exosomal *miR-203* from CRC cells can sustain the formation of the premetastatic niche by promoting the differentiation of monocytes to M2-tumor-associated macrophages (TAMs) involved in liver metastasis of CRC patients [122].

Moreover, recent evidence has suggested that liver pre-metastatic niches could be promoted by exosomal miR-21 of CRC cells, through the miR-21-*TLR7-IL6* axis [123]. Significantly increased levels of serum exosomal miR-6803-5p [124] and decreased levels of miR-548c-5p have been recently associated with poor survival and liver metastasis of CRC patients [125]. However, the molecular mechanisms by which both miR-6803-5p and miR-548c-5p influence CRC metastasis remain largely unknown. Likewise, new evidence has revealed that decreased levels of serum exosomal miR-638 are associated with an increased risk of liver metastasis [126]. Previous data demonstrated that miR-638 functions both as an oncomiR by targeting *TP53* and *PTEN* [67] and as a tumor suppressor by inhibiting *TSPAN1*, in CRC [69]. Using a translational research model, including culture cells, mouse models and human samples, Fu et al. [127] have recently demonstrated that miR-17-5p and miR-92a-3p exosomal expression is related to CRC metastasis, but not to the CRC mutational type. The area under ROC curves (AUC) of these serum exosomal miRNAs was about 0.84, while their combination leads to an AUC of 0.91. Based on a very interesting hypothesis that tumor-draining vein could provide more homogeneous information than blood drawn from the peripheral vein (PV), Monzo et al. [128] demonstrated that CRC primary tumors release higher concentrations of miRNAs through the mesenteric vein (MV) than the PV. Accordingly, they observed that exosomal miR-328 was higher in the MV than in PV plasma of the patients developing liver metastasis, suggesting a possible role of miR-328 in the development of CRC liver metastasis. However, miRNAs loading into exosomes is not a passive process. Teng et al. [129] have recently demonstrated that the sorting of oncomiRs into exosomes is suppressed, while the sorting of TS-miRs into exosomes is increased. Furthermore, they found that exosomal TS-miR including miR-193a and miR-18a are significantly higher in patients with liver metastasis compared to CRC patients having no metastasis. Functional analysis demonstrated that miR-193a modulates CRC progression by *CCDN1* and *c-MYC* expression via *Caprin1*, its direct target.

Quite interestingly, recent evidence has pointed out that miRNA-TEX released by breast cancer are directly involved in changing the normal cell phenotype into a tumor phenotype, promoting CTCs-independent metastasis, in a Dicer-dependent manner [130]. However, no data supporting this approach for CRC metastasis has been presented yet. Besides, previous data have mentioned the role of tumor-derived exosomes in the suppression of the immune system of the host [131]. Nevertheless, identifying miRNAs entrapped in exosomes released by colorectal tumors and involved in the suppression of the immune system during CRC metastasis remains a major challenge.

## 8. The Role of CRC miRNAs-TEX in the Promotion of Hepatocellular Carcinoma

The exosomes delivered by tumor cells contain both miRNAs and proteins/enzymes (Dicer, Argonaute 2 (Ago2) and TRB as well as the RISC Complex) used for miRNAs processing and silencing the mRNA targets of the recipient cells. Considering that miRNAs-TEX could modify the phenotype of the recipient cells, by modulating the cell transcriptome in a Dicer-dependent manner [130], we further analyzed the association of the CRC miRNAs-TEX mentioned above with the miRNA signature specific for the hepatocellular carcinogenesis (HCC) (Table 3).

MiR-21 is one of the most oncogenic miRNAs and its relationship with HCC has been already proved [141]. Thus, it can be speculated that the transfer of CRC miR-21-TEX into liver cells could contribute to liver metastasis. The tumor suppressor gene *BTG2* represents one of the miR-21 targets, its negative correlation with miR-21 expression being related to hepatocarcinogenesis [132]. Recent data has revealed that miR-18a has an important role in promoting liver cancer, by targeting the tumor suppressor gene *TSC1*, leading to downstream regulation of the mTOR signaling pathway [133]. More data pointed out that miR-18a is involved in liver cell proliferation by targeting *IRF2* and *CBX7* [134], as well as HCC cell migration and invasion, by inhibiting the Dicer I expression [135]. MiR-328 represents another oncomiR found into the exosomes released by CRC which would promote liver metastasis. Previous evidence highlighted the role of miR-328, as a crucial oncogene that targets and suppresses the *PTPRJ* gene, leading to an increased cellular motility in HCC [136]. Likewise, miR-17-5p, found increased in the exosomes of the metastatic CRC has an important role in liver cancer, promoting HCC cell proliferation and migration, by modulating the p38-HSP27 signaling pathway [137]. High levels of exosomal miR-92a were associated with CRC metastasis [127], but high levels of miR-92a in the liver cells have also been associated with HCC development [138]. Recent evidence has pointed out that *FOXA2* represent a direct target of miR-92a, in HCC development and progression [139]. A very interesting observation is about miR-638, which plays a dual role in colorectal tumorigenesis, modulating invasion and metastasis of CRC. Low levels in exosomal miR-638 were associated with CRC metastasis, but low levels of miR-638 were also associated with invasion and EMT in HCC, by targeting *SOX2* [140]. No data about the role of miR-548, iR-193 and miR-6803 in liver carcinogenesis has been reported yet.

## 9. Clinical Implication of miRNAs as Reliable Biomarkers

Considering their high stability in many types of biological samples [142,143], miRNAs have become important candidates for discovering new cancer biomarkers for CRC. Currently, a myriad of papers have presented the utility of serum miRNAs in CRC diagnosis, prognosis and treatment response. As an example, a set of six miRNAs including miR-21, let-7g, miR-31, miR-92a, miR-181b and miR-203 have been shown to be a reliable biomarker for CRC diagnosis with over 40% specificity and sensitivity compared to classical markers such as the carcinoembryonic antigen (CEA) and CA19-9 [144]. Another example is given by miR-200, whose increasing serum levels are significantly associated with CRC progression and liver metastasis [145]. However, a recent analysis presented by Hibner et al. [146] underline the usefulness of miRNAs as potential biomarkers for CRC diagnosis and prognosis.

Additionally, several observational studies about miRNA assessment in CRC clinical trials have been reported. One of these trials, with NCT02635087 code in the ClinicalTrials.gov Identifier, aimed to investigate whether a panel of six miRNAs (miR-21, miR-20a-5p, miR-10a-3p, miR-106b-5p, miR-143-5p and miR-215) could represent a reliable tool to predict the prognosis of the patients with stage II CRC treated with chemotherapy. The miRNAs expression will be evaluated from about 630 surgical tissue samples, and the final results are estimated to be published in July 2020. Another trial, coded NCT03362684 in ClinicalTrials.gov identifier, aimed to establish if the expression of the miR-31-3p and miR-31-5p could be used as prognostic factors for patient outcomes or predict the benefit from anti-EGFR therapy in stage III Colon Cancer. Although this study ended in June 2016, there are no published results yet. New data about the clinical role of miRNA in CRC could come from the trial NCT03309722 initiated in October 2008 by the University of Southampton from UK that aimed to investigate the role of several miRNAs and their targets in colorectal cancer. This observational study does not mention which are the specific miRNAs studied and no results are published yet. The primary outcome measures will include overall survival, disease-free survival, local recurrence and distant recurrence, all evaluated at five years. Estimated enrollment for this study is expected to be around 1000 participants and first results are waited to be published in October 2020.

## 10. Conclusions

Growing evidence consistently shows that miRNAs represent important molecules in the modulation of all hallmarks of cancer. Accordingly, specific miRNAs have been associated with the diagnosis, development, and progression of CRC as well as its liver metastasis. Moreover, changes in the circulating miRNAs levels in patients with solid cancers could be considered for developing novel diagnostic and prognostic factors. In this regard, blood miRNAs transported by CRC-delivered exosomes (miRNA-TEX) have become valuable tools that could be used to predict CRC metastasis in the liver. Several CRC oncomiRs, including miR-18a, mir-328, miR-17-5p and miR-92a, delivered into tumor exosomes, are associated with CRC metastasis, but have also been associated with liver carcinogenesis. However, before their implementing in the clinic as reliable biomarkers, more validation studies are necessary, while standardized methods related to miRNAs processing, expression, and normalization have to be established.

As future challenges, we consider that it would be of interest to evaluate how much of the liver miRNAs could be altered by the chemotherapy regimens used to treat the primary CRC. In the case that these alterations exist, the question arises if and how these miRNAs could be involved in facilitating CRC metastasis in the liver.

## Figures and Tables

**Figure 1 ijms-19-03711-f001:**
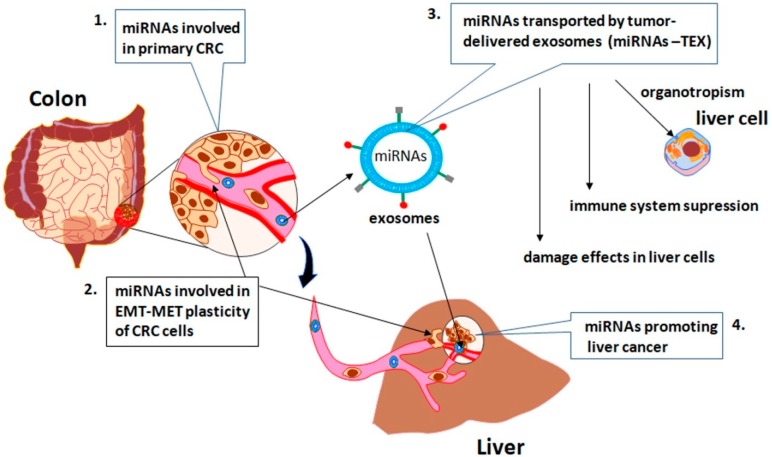
The PubMed string searches used to identify the specific miRNAs involved in colorectal cancer (CRC) development and progression as well as in liver metastasis.

**Figure 2 ijms-19-03711-f002:**
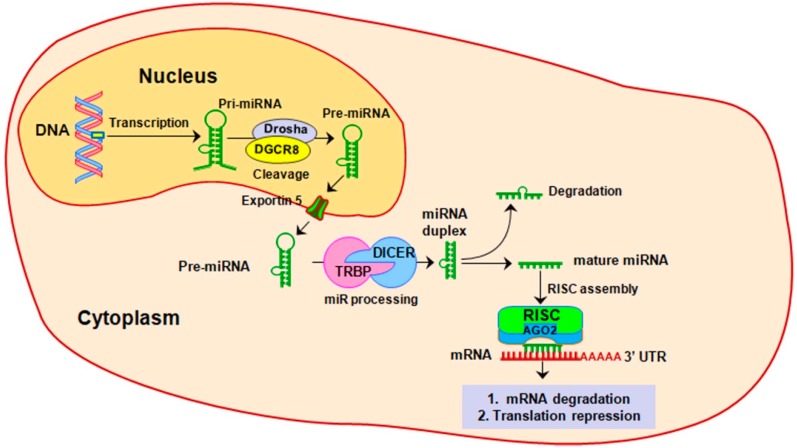
MiRNA biogenesis. MiRNA biogenesis starts in the nucleus with a long hairpin transcript called pri-miRNA that is further processed to a smaller transcript of 70 nucleotides. After is exported in the cytoplasm, pre-miRNA is enzymatically processed to single-strand mature miRNA of about 21–23 nucleotides. When it is incorporated in the enzymatically machinery called the RNA-induced silencing complex (RISC), mature miRNA guide the RISC to silence the specific mRNA transcripts, either by their degradation of by translation repression.

**Figure 3 ijms-19-03711-f003:**
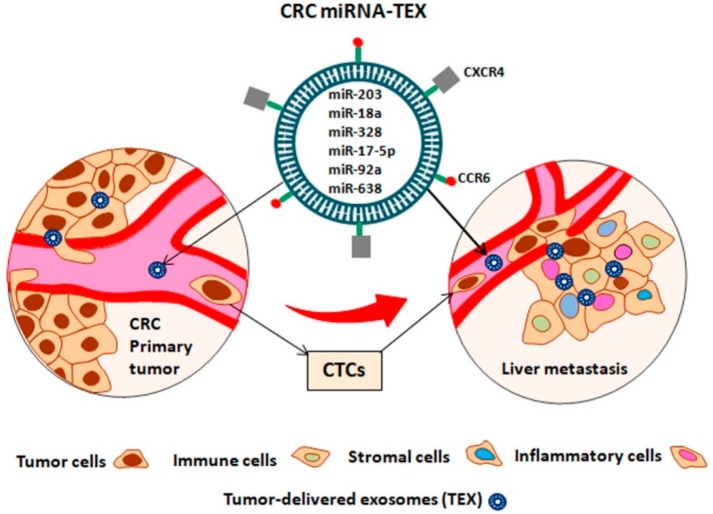
CRC metastasis in the liver through CRC circulating tumor cells (CTCs), mediated by miRNAs transported by CRC-delivered exosomes (miRNA-TEX). *CXCR4* and *CCR6* chemokine receptors are responsible for liver organotropism of miRNA-TEX, while the miRNAs released in the liver are responsible for preparing the secondary pre-metastatic niche by inducing the pro-inflammatory and pro-tumorigenic signals.

**Table 1 ijms-19-03711-t001:** MiRNAs and their mRNA targets associated with the development, progression and metastasis of CRC.

Key Signaling Pathways	miRNAs	Targets	Targeting Effects	Ref.
Activation of Wnt/ß-catenin pathway	**oncomiR (High Expression)—Gain of Function**			
	miR-21	*CTNNB1*, *TGFBR2*, *PIK3CA*, *BRAF*, *ZFHX3*, *SFRP1*	Tumor development, proliferation, progression	[30,31]
	miR-135a/b	*APC*	proliferation	[32]
	miR-494	*APC*	Proliferation, tumorigenesis	[33]
	miR-155	*CTNNB1*	Invasion; metastasis	[34]
	miR-146a	*NUMB*	Progression, stemness	[35]
	miR-522	*DACH1*	Proliferation, migration	[36]
	miR-19a	*APC*, *CTNNB1*, *c-Myc*, *PTEN*, *TIA1*	Proliferation, tumorigenesis, proliferation, invasion, progression, angiogenesis	[37,38]
	miR-224	*GSK3β*, *sFRP-2*	Proliferation, metastasis	[39]
	miR-103a miR-1827	*APC*, *APC2*, *CTNNB1*, *WNT3a*	Cell cycle progression, reduced apoptosis	[40]
	miR-574-5p	*Qki 6/7*	Proliferation, tumorigenesis differentiation, angiogenesis	[41]
	**TS-miR (Low/Reduced Expression)—Loss of Function**			
	miR-137	*CTNNB1*, *WNT3a*	Cell cycle progression	[40]
	miR-23b	*FzD-7*	Proliferation, progression, invasion, metastasis	[42]
	miR-7	*YY-1*	Proliferation, reduced apoptosis, cell cycle progression	[43]
	miR-93	*SMAD-7*, *CTNNB1*	Proliferation, progression	[44]
	miR-185	*MYC*, *CCND1*	Proliferation, progression	[45]
Activation of EGFR signaling pathway	**oncomiR (High Expression)—Gain of Function**			
	miR-20, miR-21, miR-130b	*PTEN*	Progression, invasion, metastasis	[46]
	miR-26b	*PTEN*, *WNT5A*	Proliferation, EMT, metastasis	[47]
	miR-182, miR-135b	*ST6GALNAC2*, *PI3K/AKT*	Proliferation, invasion	[48]
	**TS-miR (Low/Reduced Expression)—Loss of Function**			
	miR-43, miR-145	*CD44*, *KLF5*, *KRAS*, *BRAF*	Proliferation, migration	[49]
	Let-7	*KRAS*	tumorigenesis	[50]
	miR-19a	*KRAS*, *VEGFA*	Proliferation, angiogenesis	[51]
	miR-181d	*PEAK1*	Proliferation, invasion, migration, metastasis	[52]
	miR-30a	*KRAS*, *ME1*	CRC developmnet	[53]
	miR-217	*MAPK1*, *KRAS*, *Raf-1*	Tumor growth, apoptosis	[54]
	miR-487b	*LRP6*, *KRAS*	Metastasis	[55]
	miR-16	*KRAS*	Proliferation, invasion, apoptosis	[56]
	miR-384	*KRAS*, *CDC42*	Invasion, migratiuon, metastasis	[57]
	mirR-125a-3p	*FUT5-FUT6*	Proliferation, migration, invasion, angiogenesis	[58]
Inactivation of TGF-ß signaling pathway	**oncomiR (High Expression)—Gain of Function**			
	miR-135b	*TGFBR2*	Progression, inhibiting of apoptosis	[59]
	miR-301a	*TGFBR2*	Migration, invasion, metastasis	[60]
	miR-20-5p	*SMAD4*	EMT, Invasion, metastasis	[61]
	miR-224	*SMAD4*	Invasion, metastasis	[62]
Suppressing of TP53 function	**oncomiR (High Expression)—Gain of Function**			
	miR-125b	*TP53*	CRC progression	[63]
	miR-504	*TP53*	CRC progression	[64]
	miR-29b, miR-30	*TP53*	CRC progression	[65]
	miR-24	*TP53*	CRC pogressin	[66]
	miR-518c, miR-638	*TP53*, *PTEN*	CRC progression, invasion, metastasis	[67]
	**TS-miR (Low/Reduced Expression)—Loss of Function**			
	miR-638	*SOX2*, *TSPAN1*	EMT, invasion, migration, proliferation	[68,69]
	miR-34a	*SIRT1*	Proliferation, reducing apoptosis	[70]

**Table 2 ijms-19-03711-t002:** MiRNAs and their mRNA targets associated with regulation and induction of epithelial to mesenchymal transition (EMT) and mesenchymal-epithelial transition (MET), supporting CRC invasion and metastasis.

Key Signaling	miRNAs	Targets	Function	Ref.
Regulation of EMT	**oncomiR (High Expression)—Gain of Function**			
	miR-675-5p	*SNAIL*	EMT, invasion, metastasis	[83]
	**TS-miR (Low/Reduced Expression)—Loss of Function**			
	miR-34a/b/c	*SNAIL*, *SLYG*, *ZEB1*, *BIM1*, *CD44*, *CD133*, *c-MYC*	EMT, invasion, metastasis, MET	[84,85]
	miR-375	*SP1*, *MMP2*, *SNAIL*, *CDH1*, *VIM*, *CDH2*, *CTNNB1*	EMT, invasion, metastasis	[86]
	miR-374	*CCND1*, *ZEB1*, *CDH2 VIM*, *SLUG*, *SNAIL*	EMT, proliferation, invasion, migration, liver metastasis	[87]
	miR-200c	*ZEB1*, *ETS1*, *FLT1*, *ASCL2*	EMT-MET plasticity, invasion, migration, liver metastasis	[88,89]
	miR-429	*ONECUT2*, *ZEB1*, *ZEB2*	EMT, invasion, migration	[90]
	miR-335, miR-132, miR-192	*ZEB2*	EMT, Invasion, metastasis	[91,92,93]
Inducing of EMT	**oncomiR (High Expression)—Gain of Function**			
	miR-20a	*SMAD4*, *MMP2*, *MMP9*	EMT, migration, metastasis	[94,95]
	miR-4775	*SMAD7/TGF-ß*	EMT, invasion, metastasis	[96]
	miR-1269	*SMAD7*, *HOXD10*	EMT, invasion, metastasis	[97]
	miR-21	*NRF2*, *SMAD7*	EMT, invasion, metastasis	[98]
	miR-150	*CREB*	EMT, invasion, migration	[99]
	miR-194	*MMP2*	EMT, invasion, migration	[100]
	**TS-miR (Low/Reduced Expression)—Loss of Function**			
	miR-320c	*SOX4*, *FOXM1*, *FOXQ1*	EMT, proliferation, migration, tumorigenesis	[101]
	miR-187	*SOX4*, *NT5E*, *PTK6*	EMT, invasion, metastasis	[102]
	miR-34a	*CTNNB1*, *c-Met ZEB1*, *ZEB2*, *SNAIL*	Progression, EMT, liver metastasis	[103,104]
	miR-145	*SIP1*, *CTNNB1*, *TCF4*, *VIM*, *SNAIL*	Proliferation, migration, EMT, invasion, metastasis	[105]
	miR-29b	*BCL9L*, *TCF7L2*, *SNAI1*, *PGRN*	Cell growth, EMT, angiogenesis, invasion, migration	[106,107]
	Let-7c	*MMP11*	Cell migration and invasion	[108]

**Table 3 ijms-19-03711-t003:** Common miRNAs identified both in exosomes-delivered by CRC cells and in the early diagnosis of hepatocellular carcinomas.

CRC miRNAs–TEX	HCC Targets	Function	Ref.
**oncomiR**			
Mir-21	*BTG2*	Tumorigenesis	[132]
Mir-18a	*TSC1*	Tumorigenesis	[133]
	*IRF2*, *CBX7*	Cell proliferation	[134]
	*Dicer I*	Migration and invasion	[135]
miR-328	*PTPRJ*	Tumor progression, motility	[136]
miR-17-5p	*P38*, *HSP27*	Proliferation and cell migration	[137]
mir-92a	*FOXA2*	Inhibiting apoptosis, tumorigenesis	[138,139]
**TS-miR**			
miR-638	*SOX2*	EMT, development	[140]

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
