# Peer review of "The Impact of miRNA in Colorectal Cancer Progression and Its Liver Metastases"

_ijms, 2018, doi:10.3390/ijms19123711_

Reviewer 1 Report

The impact of miRNA in colorectal cancer progression and its liver metastases

Balacescu, Sur D, Cainap C, Visan S, Cruceriu D, Manzat-Saplacan R, Muresan MS, Balacescu L, Lisencu C, Irimie A

The paper provides a systemic collection of information regarding miRNAs role in colorectal   progression, diagnosis and liver metastases. The authors have explained specific miRNAs involved in Wnt/Bcatenin, EGFR, TGF beta signaling pathways. The role of miRNA in regulation of epithelial-mesenchymal associated genes resulting in CRC metastases has been fully discussed. Furthermore, based on the recent studies, they have elaborated how miRNA transported by CRC-delivered exosomes can facilitate and sustain CRC metastases to the liver and promote hepatocellular carcinoma, besides, these tumor delivered exosomes could be used for prediction of early diagnosis of hepatocellar carcinoma.

PROS

The paper is well written and has a good flow.

The authors have performed a comprehensive search on miRNA and related pathways and most of the recent studies on this subject has been covered in this review.

Classification of specific miRNAs involved in different pathways in separate tables provides a handy resource for researchers.

Cons

The authors could extend their study by including some information regarding clinical implication of miRNAs as reliable biomarkers.

Author Response

Dear Reviewer, 

We thank you for your pertinent observations regarding our study and we consider that addressing your request we improved the quality of our manuscript entitled “The impact of miRNA in colorectal cancer progression and its liver metastases”.

Please find attached our answer to your request.

Reviewer 2 Report

            The manuscript entitled “The impact of miRNA in colorectal cancer progression and its liver metastases” by Balacescu et al. reviews the participation of miRNAs in colorectal cancer (CRC). Since the discovery of miRNAs as epigenetic modulators of gene expression, a plethora of papers on their occurrence and functionality have appeared. The role of miRNA in cancer onset, development and progression has also been extensively studied. Among other types of cancer, CRC is influenced by miRNA deregulation and this issue has also been the subject of numerous reviews.

            In spite of this abundance of data available in the literature, the manuscript by Balacescu et al. exhibits some valuable features. First, it systematically reviews the participation of miRNAs in primary tumour development. The authors classify the different miRNAs according to the pathways in which they are involved. Then go on to describe the involvement of miRNAs in the epithelial to mesenchymal transition (EMT), their transport via exosomes to other organs, especially liver, and their contribution to the preparation of pre-metastatic niches, to finally deal briefly with development of secondary tumours. A second merit of the present manuscript is that the literature has been extensively reviewed, paying special attention to the last reports in the field. In fact, one third of the references given correspond to 2017 and 2018. Moreover, the authors include a section (lines 63 ff.) devoted to the method used to search the literature. This is a valuable detail that is overlooked in most review articles.

            The manuscript is well written, and the tables included in it facilitate the identification of the targets of the different miRNAs. The different sections are in general preceded by a brief introduction explaining the biological foundations of the processes involved. To quote a single example, the nature of exosomes and their role in preparing the pre-metastatic niche are briefly and yet clearly exposed at the beginning of section 6.

            There are, however, some details that may limit the value of the manuscript. They are listed below, as they may require a modification of the text.  

1) While, as above mentioned, the biological foundations of the different processes are adequately described, a brief introduction on the mechanisms involved in miRNAs function is missing. This is especially important; the potential readers may come from very different fields: medical oncology, molecular biology, epigenetics, etc. and not all of them may be acquainted with the role of miRNAs. Including a brief introduction in this sense may increase the visibility and comprehension of the article, especially when the role of Dicer-dependent mechanisms is mentioned in the establishment of metastases.

2) Although not essential for the comprehension of the review, a brief reference to the methods used to predict the miRNAs targets would illustrate the importance of the research with these non-coding RNAs. Many bioinformatic procedures exist to do it, and their relative value has been recently reviewed in the Int. J. Mol. Sci.

3) In some parts of the manuscript (e.g. line 106) the regulator of miRNAs expression is given. The authors may decide on the convenience of including these details, when available, in the tables, as this inclusion would improve the information recorded.

4) Ideally, a review article, apart of describing the pertinent original results, should provide a brief prospect of the future developments in the field. This may aid the potential readers working in that area to plan their research. This question is missing in the present manuscript and its inclusion would be desirable.

Author Response

Dear Reviewer, 

We thank you for your pertinent observations regarding our study and we consider that addressing your requests we improved the quality of our manuscript entitled “The impact of miRNA in colorectal cancer progression and its liver metastases”.

Please find attached our answer to your request.
